# Non-invasive imaging of CSF-mediated brain clearance pathways via assessment of perivascular fluid movement with diffusion tensor MRI

Ian F Harrison[1], Bernard Siow[1,2], Aisha B Akilo[1], Phoebe G Evans[1], Ozama Ismail[1], Yolanda Ohene[1], Payam Nahavandi[1], David L Thomas[3,4], Mark F Lythgoe[1], Jack A Wells[1]*

[1]UCL Centre for Advanced Biomedical Imaging, Division of Medicine, University College London, London, United Kingdom; [2]The Francis Crick Institute, London, United Kingdom; [3]Neuroradiological Academic Unit, Department of Brain Repair and Rehabilitation, UCL Institute of Neurology, London, United Kingdom; [4]Leonard Wolfson Experimental Neurology Centre, UCL Institute of Neurology, London, United Kingdom

**Abstract** The glymphatics system describes a CSF-mediated clearance pathway for the removal of potentially harmful molecules, such as amyloid beta, from the brain. As such, its components may represent new therapeutic targets to alleviate aberrant protein accumulation that defines the most prevalent neurodegenerative conditions. Currently, however, the absence of any non-invasive measurement technique prohibits detailed understanding of glymphatic function in the human brain and in turn, it's role in pathology. Here, we present the first non-invasive technique for the assessment of glymphatic inflow by using an ultra-long echo time, low *b*-value, multi-direction diffusion weighted MRI sequence to assess perivascular fluid movement (which represents a critical component of the glymphatic pathway) in the rat brain. This novel, quantitative and non-invasive approach may represent a valuable biomarker of CSF-mediated brain clearance, working towards the clinical need for reliable and early diagnostic indicators of neurodegenerative conditions such as Alzheimer's disease.

DOI: https://doi.org/10.7554/eLife.34028.001

*For correspondence:
jack.wells@ucl.ac.uk

**Competing interests:** The authors declare that no competing interests exist.

**Reviewing editor:** Thomas W Okell,

## Introduction

The recent identification of the glymphatic system and the dural lymphatic network provide exciting new perspectives on waste clearance mechanisms within the central nervous system (CNS) (*Louveau et al., 2015*; *Iliff et al., 2012*). According to the glymphatics hypothesis, cerebrospinal fluid (CSF) crosses from the subarachnoid space into the periarterial space where it swiftly flows towards the brain tissue. Fluid then passes into the parenchyma from the perivascular space, a transition mediated by aquaporin-4 (AQP4) channels that reside on the end feet of astrocytes. This periarterial inflow creates a convective flux of fluid across the parenchyma that exits via perivenous channels, carrying with it 'waste products' of brain metabolism. As such, the glymphatic pathway has been proposed to function as a 'cleaning system' of the brain. The exchange of CSF with interstitial fluid (ISF) is an established mechanism underlying the clearance of amyloid beta (Aβ), recognised as a leading molecular candidate to initiate Alzheimer's disease (AD) (*Iliff et al., 2012*; *Weller et al., 2008*; *Bakker et al., 2016*; *Xu et al., 2015*; *Tarasoff-Conway et al., 2015*; *Hawkes et al., 2014*).

**eLife digest** Our brain is bathed in cerebrospinal fluid, a clear liquid that 'cushions' the fragile organ. This liquid travels into the brain along special channels – the perivascular space – that surround certain blood vessels. As the fluid washes in and out of the brain, it takes with it potentially harmful molecules, such as the aggregates that build up to cause Alzheimer's disease. If this brain-cleaning system becomes faulty, it could lead to neurodegenerative diseases. However, it is extremely difficult to measure the activity of this intricate and delicate system, and most studies so far have had to use invasive techniques that usually require brain surgery.

Now, Harrison et al. adapt a technique, called diffusion tensor magnetic resonance imaging (MRI), to visualise how the cerebrospinal fluid moves in the perivascular space in healthy rats. The non-invasive MRI method captures how the cerebrospinal fluid is driven into the brain when the blood vessels nearby expand and contract; as the vessels pulsate with each heartbeat, there is a 300% increase in the movement of the fluid in the perivascular space.

This approach could be applied to understand exactly how neurodegenerative diseases emerge when the cerebrospinal fluid stops to properly clean the brain. Ultimately, the method could be used to detect when the cleansing system starts to fail in people, which could help to treat patients before their brains accumulate too many harmful substances.

DOI: https://doi.org/10.7554/eLife.34028.002

Despite evidence that aspects of the glymphatic pathway are preserved across species (*Goulay et al., 2017*; *Ringstad et al., 2017*; *Dobson et al., 2017*), key questions remain on the anatomy and function in the human brain and to what extent it contributes to pathology. Currently, however, these questions cannot be answered because there are no non-invasive techniques for assessment. The development of non-invasive methods to image CSF-mediated brain clearance pathways, such as the glymphatic system, would enable repeated and practical measurement to investigate this system in the human brain and the intact animal skull. This, in turn, may help fully characterise impairment of CSF-mediated clearance pathways with age (*Kress et al., 2014*), as well as the contribution to Aβ accumulation in AD. Ultimately, such methods could address the pressing clinical need for reliable and early biomarkers of AD, by identifying patients at risk of Aβ accumulation due to failing clearance mechanisms.

The perivascular space is a fluid filled compartment that surrounds selected blood vessels in the brain (*Huffman et al., 2016*). Perivascular channels form a central component of the glymphatic pathway that is said to drive rapid CSF-ISF exchange. Although the precise routes and fluid dynamics that underlie CSF-ISF exchange remain controversial (*Holter et al., 2017*; *Hladky and Barrand, 2014*; *Brinker et al., 2014*; *Smith et al., 2017*), several independent groups have identified perivascular channels as central to this pathway (*Iliff et al., 2012*; *Bedussi et al., 2015*; *Lochhead et al., 2015*; *Rennels et al., 1985*). As such, the perivascular space represents a promising target for non-invasive imaging biomarkers of CSF-ISF exchange. To date, perivascular function has been studied using only invasive methods: ex-vivo microscopy (*Bedussi et al., 2015*), two-photon imaging (*Iliff et al., 2012*) and contrast-enhanced MRI following intra-cranial/lumbar injection (*Iliff et al., 2013a*; *Yang et al., 2013*). In this work we introduce the first non-invasive method for the assessment of perivascular function using contrast-free MRI, and demonstrate use of the method in the rodent brain.

Despite recognition that the perivascular space facilitates CSF-ISF exchange, the nature of fluid movement within this channel is yet to be unambiguously determined. Broadly, the glymphatics hypothesis describes perivascular fluid movement as possessing coherent, bulk flow (*Iliff et al., 2012*). However, this has been questioned by other studies which propose that the fast distribution of CSF-tracers along the perivascular space can be explained by rapid dispersion of fluid/tracers via mechanical pulsations, with little bulk flow (*Hladky and Barrand, 2014*; *Asgari et al., 2016*). Given the current uncertainty, when considering non-invasive MRI techniques for assessment, diffusion MRI represents a prime candidate for initial application owing to its established sensitivity to water dispersion, together with evidence of sensitivity to bulk flow (non-plug e.g. laminar flow) from prior studies of the cerebral vasculature (*Wells et al., 2017*). That is, irrespective of whether perivascular fluid movement is dominated by bulk flow or rapid dispersion with little bulk flow, diffusion MRI

sequences, if appropriately tuned, should yield sensitive and quantitative correlates of fluid movement, albeit non-specific to flow coherence.

In this study, we apply ultra-long echo time (TE), diffusion weighted MRI sequences to assess fluid movement within perivascular channels surrounding the middle cerebral artery (MCA) of the healthy rat brain. In addition, given evidence that cerebral arterial pulsation is a key mechanism that drives PVS fluid movement (*Rennels et al., 1985*; *Iliff et al., 2013b*), we investigate the dependence of the technique on vascular pulsatility through cardiac gating and modulation by the adrenoceptor agonist, dobutamine. This technique represents the first non-invasive biomarker of perivascular action, working towards new translational techniques to assess CSF mediated brain clearance pathways and their role in disease.

## Results

### Non-Invasive imaging of perivascular channels

The ultra-long TE MRI sequence presented here is designed to attenuate the measured signal from the blood and parenchyma that immediately surround the perivascular space in order to minimise partial volume effects, which represent a potential confounder for assessment by MRI given the small size of this compartment. *Figure 1A* shows a b0 image of the axial slice through the ventral aspect of the rat brain. The subarachnoid CSF that bathes the Circle of Willis (CoW) can be clearly observed, with marked contrast between the blood vessels within the CoW and surrounding CSF. Bright tracts appear either side of both MCA branches (*Figure 1A*) which, due to the ultra-long echo time, must derive from fluid filled compartments of similar composition to the CSF in the subarachnoid space. This observation, together with the characteristic morphology that runs alongside and parallel to the MCA, is consistent with the description of the perivascular space as a fluid filled compartment that surrounds major blood vessels feeding the brain (*Huffman et al., 2016*). Indeed, the location of this compartment is highly consistent with direct assessment from a previous study (*Figure 1B*, adapted from Lochhead *et al.*, [*Lochhead et al., 2015*]).

The precise definition of the perivascular (and 'paravascular') space is somewhat unclear, as highlighted in a number of recent articles (*Hladky and Barrand, 2014*; *Brinker et al., 2014*; *Bedussi et al., 2017*). Whether the fluid filled tracts around the MCA that we observe (*Figure 1A*) occupy a physically and functionally distinct 'paravascular' space as described by Iliff *et al.*, (*Iliff et al., 2012*) forms a more continuous pathway with subarachnoid CSF as described by Bedussi *et al* (*Bedussi et al., 2017*)., or are well described by a perivascular space as proposed by Lochhead *et al.*, (*Lochhead et al., 2015*) remains unknown. Irrespective of the precise anatomical bordering of the fluid filled tracts identified in this work, and despite these semantic differences, all the aforementioned studies have highlighted the movement of fluid that surrounds subarachnoid arteries as a key site of CSF-tracer inflow towards the parenchyma. Hence non-invasive assessment of fluid movement within this compartment represents a meaningful measure of CSF-ISF exchange pathway function.

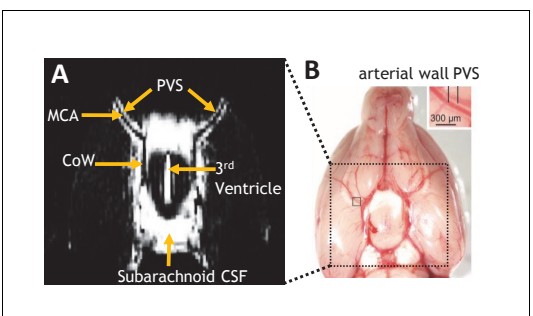

**Figure 1.** Non-invasive MRI of perivascular channels. (**A**) Example b0 MRI image. The position and orientation of the imaging slice is adjusted to optimally visualize the perivascular space (PVS) around both branches of the MCA. Bright signal can be observed from fluid filled compartments: CSF in the subarachnoid space around the Circle of Willis (CoW); fluid in the perivascular space that surrounds the MCA; the ventral aspect of the third ventricle. (**B**) Photograph of the ventral aspect of the rat brain surface illustrating a putative PVS surrounding the middle cerebral artery (MCA) (reproduced with permission from *Lochhead et al., 2015*).

DOI: https://doi.org/10.7554/eLife.34028.003

### Assessment of fluid movement using Multi-Direction diffusion weighted imaging

Application of a motion probing gradient (MPG) along the principle direction of the perivascular tracts located around the MCA was observed to

markedly attenuate the signal from these tracts relative to when the MPG was applied perpendicular to their principle orientation (*Figure 2A*). Accordingly, across the 10 subjects, within the right perivascular space, the pseudo-diffusion coefficient (D*) parallel to PVS orientation was significantly greater than D* in either perpendicular direction ($p<0.01$ respectively). In a similar fashion, D* (parallel to principle direction of left PVS) was significantly greater than D* in either perpendicular direction [$p<0.01$]. (*Figure 2B*). These data demonstrate that the MRI sequence employed here can detect the directional dependence of fluid movement within the perivascular space (the principal directionality of which is parallel to their orientation), which verifies that they are sensitised to the movement of fluid within this compartment. Within the CSF in the subarachnoid space, it was observed that D* when the MPGs were applied in the in-plane orientation (i.e. parallel to the left or right branch of the MCA) were both significantly greater than D* in the through plane orientation [$p<0.01$]. This is consistent with the known direction of CSF movement in the rostral-caudal direction within this region from prior invasive studies (*Iliff et al., 2013a*; *Lee et al., 2018*).

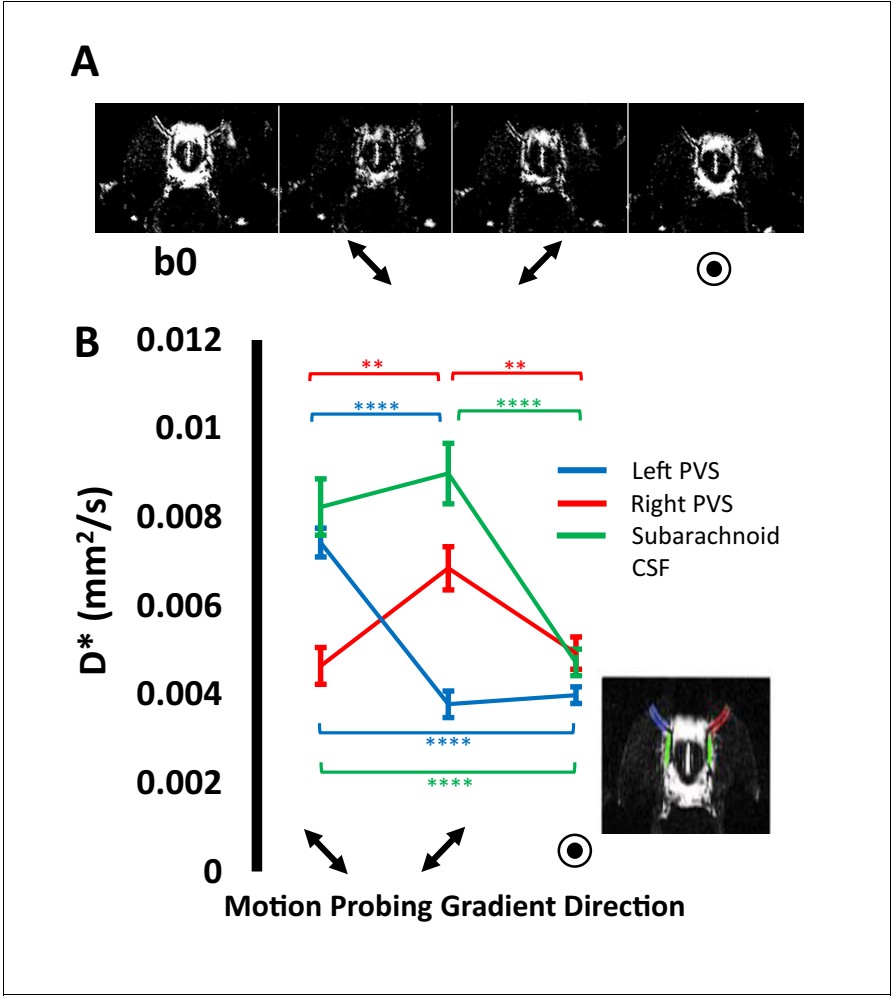

**Figure 2.** Multiple direction diffusion weighted imaging of subarachnoid and perivasular fluid movement. (**A**) Example b0 and 'diffusion weighted' images acquired with the motion probing gradients applied in 3 orthogonal directions respectively. (**B**) The mean D* calculated within ROIs [see insert] in the right perivascular space (red), left perivascular space (blue), subarachnoid space (green) with the motion probing gradients applied in three orthogonal directions (±SEM).
DOI: https://doi.org/10.7554/eLife.34028.004
The following figure supplement is available for figure 2:

**Figure supplement 1.** The individual animal D* calculated within ROIs in the right perivascular space.
DOI: https://doi.org/10.7554/eLife.34028.005

## Diffusion tensor imaging of CSF/Perivascular Fluid Movement

Having verified the sensitivity of the MRI sequence to fluid movement within the perivascular and subarachnoid space, MPGs were then applied in six different directions to generate a pseudo diffusion tensor image that reflects the directionality and magnitude of subarachnoid CSF and perivascular fluid movement.

*Figure 3* illustrates that, for the subarachnoid space ROI, the mean D* tensor ellipsoid (n = 6) was well aligned with the known principle direction of CSF movement (caudal-rostral, observed in several invasive studies of the rodent brain [*Iliff et al., 2013a*; *Lee et al., 2018*]). Likewise, *Figure 3* illustrates that the principle direction of the mean D* tensor of the left and right perivascular space, respectively, was aligned with the orientation of the respective branch of the MCA. The D* tensors for each of the individual animals are shown in *Figure 2—figure supplement 1*, which show reasonable consistency with the directionality of the mean tensors shown in *Figure 3*. The magnitude of the D* tensors within this region were markedly reduced post-mortem, which demonstrates that a large component of the D* measurements reflects fluid movement driven by physiological perturbations such as cardiac and respiratory pulsation and secretion from the choroid plexus (*Figure 3—figure supplement 1*). This may also partially reflect the reported collapse of the PVS post mortem [1] (indeed visual inspection of the b0 images indicates a reduction in signal intensity within this region [data not shown]). Fractional anisotropy (SEM) within the right and left perivascular space and the subarachnoid space was 0.44 (±0.04), 0.36 (±0.04) and 0.6 (±0.02) respectively with mean diffusivity (SEM) calculated to be 0.0042 (±0.0003), 0.0052 (±0.0003), 0.0065 (±0.0007) mm$^2$/s. *Figure 3E* shows a map of pseudo diffusion tensors for a single subject. The principal direction of the D* tensors in the perivascular tracts that surround the left and right MCA respectively can be seen to run parallel to the orientation of the MCA. Likewise, the principal orientation of the individual voxel D* tensors can be seen to run rostral-caudal in the mid-section of the CoW.

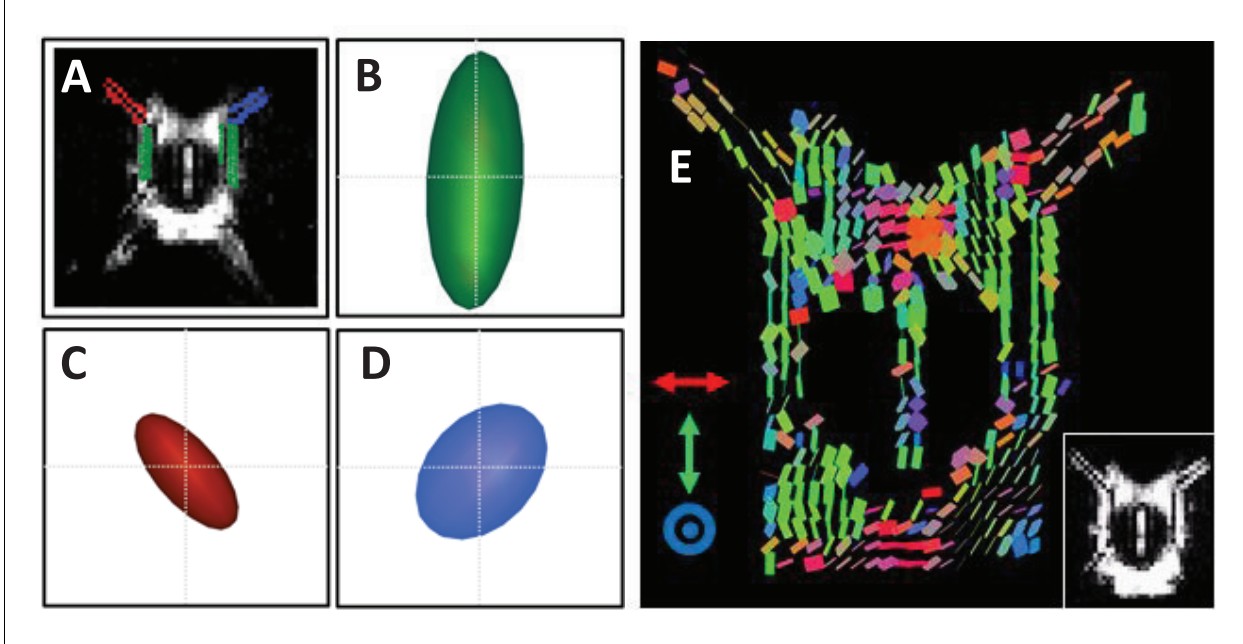

**Figure 3.** Diffusion tensor imaging of subarachnoid and perivascular fluid movement. (**A**) bO image with ROIs in the right and left PVS and subarachnoid space highlighted in blue, red and green respectively. The mean pseudo-diffusion tensor ellipsoid within the subarachnoid space ROI (**B**) and right (**C**) and left (**D**) PVS respectively across the six rats. The pseudo-diffusion tensors for each individual animal are shown in *Figure 3—figure supplement 1*. (**E**) Example map of pseudo-diffusion tensor ellipsoids with corresponding b0 image (insert).
DOI: https://doi.org/10.7554/eLife.34028.006

The following figure supplement is available for figure 3:

**Figure supplement 1.** D* tensors within the left and right perivascular space (PVS) and subarachnoid space ROIs for each of the individual subjects imaged in part ii).
DOI: https://doi.org/10.7554/eLife.34028.007

# Cerebral arterial pulsation drives Non-invasive measures of perivascular fluid movement

Previous studies have identified cerebral vascular pulsation to play a prominent role in perivascular fluid propulsion. To investigate this mechanism, MRI data were captured during both cerebral arterial pulsation and diastole using ECG gating with variable delays to image capture (36 ms and 116 ms from the r-wave to the centre of 'diffusion' weighting respectively). The results are shown in *Figure 4*, where a striking and highly directional dependence of D* on cerebral vascular pulsation was observed in the PVS [*Figure 4*]. D* in the PVS was ~300% greater during arterial pulsation relative to diastole when motion probing gradients were applied parallel to the principle orientation (p<0.01). We recorded a more moderate dependence (p=0.1) on the r-wave delay within the CSF ROI at the mid-section of the CoW (although visual inspection of the D* maps suggests that other regions within the subarachnoid CSF appeared to show greater changes with the r-wave delay). Minimal dependence of the D* measures on the r-wave delay was observed in the third ventricle (p=0.2).

Administration of the adrenoceptor agonist, dobutamine, increased heart rate from (354 ± 8 to 519 ± 17 bpm). A 65% increase in D* along PV channels was recorded (p<0.01) following dobutamine with comparatively little change after vehicle (*Figure 4C*). No significant changes were observed in the subarachnoid space ROI at the mid-section of the CoW following dobutamine (p=0.39, although visual inspection of the data suggests other regions within the subarachnoid CSF did show marked increases in D*). Dobutamine had minimal effect on D* within the third ventricle (p=0.30).

Together these data are concordant with previous invasive measures demonstrating that perivascular fluid movement is driven by cerebral vascular pulsation and that we are now able to capture this mechanistic dependence non-invasively using the techniques introduced here.

## Discussion

In this study, we introduce a novel MRI method to measure a distinct feature of brain physiology that, to date, has only be assessed using invasive methods – the movement of fluid in the perivascular space. The perivascular space serves as a preferential pathway for CSF-ISF exchange, an important mechanism supporting the clearance of potentially harmful molecules, such as Aβ, from the CNS. This non-invasive and translational method may have utility in AD research given evidence that Aβ accumulation (in late stage, sporadic AD) occurs not because of increased Aβ production but because of decreased rates of Aβ clearance (*Mawuenyega et al., 2010*). Thus, this technique may expedite greater understanding of how Aβ clearance mechanisms become impaired with ageing (*Kress et al., 2014*) and in turn reveal a new window in early AD pathogenesis in which to target future diagnostic and treatment strategies. The technique may have broader utility to a range of neurological conditions given reported associations between glymphatic function in, for example, stroke (*Gaberel et al., 2014*) and traumatic brain injury (*Iliff et al., 2014*).

The precise mechanisms that underlie CSF-ISF exchange are yet to be fully defined and this remains an active area of research. Accumulative evidence, however, has established cerebral vascular pulsation as an important mechanism underlying perivascular fluid movement (*Rennels et al., 1985*; *Iliff et al., 2013b*). Here, we have captured the action of cerebral arterial pulsation to drive perivascular fluid movement using non-invasive techniques (*Figure 4*). The measured D* showed a remarkable dependence on vascular pulsation with a ~300% increase recorded during arterial pulsation relative to diastole (*Figure 4*). Moreover, D* (non-gated) was found to markedly increase following adrenoceptor agonist, dobutamine. The non-invasive nature of this technique may enable future studies to investigate the mechanistic link between vascular pulsatility and PVS fluid movement in the healthy human brain, and its modulation by pathology as well as novel therapy.

In this study, D* estimates were captured using a b0 image and then with motion probing gradients applied at a single *b*-value, in different directions. Future studies may wish to examine the behaviour of the PVS signal over a greater range of b-values (and different values of δ and Δ) to examine whether, in combination with more advanced signal modelling, this may reveal more detailed insight into PVS fluid movement. Of note, a previous study aimed to correlate MRI measures of water diffusivity from the PVS to AD severity (*Taoka et al., 2017*). However, this earlier work presents limited evidence as to the contribution of the perivascular space to the measured MRI

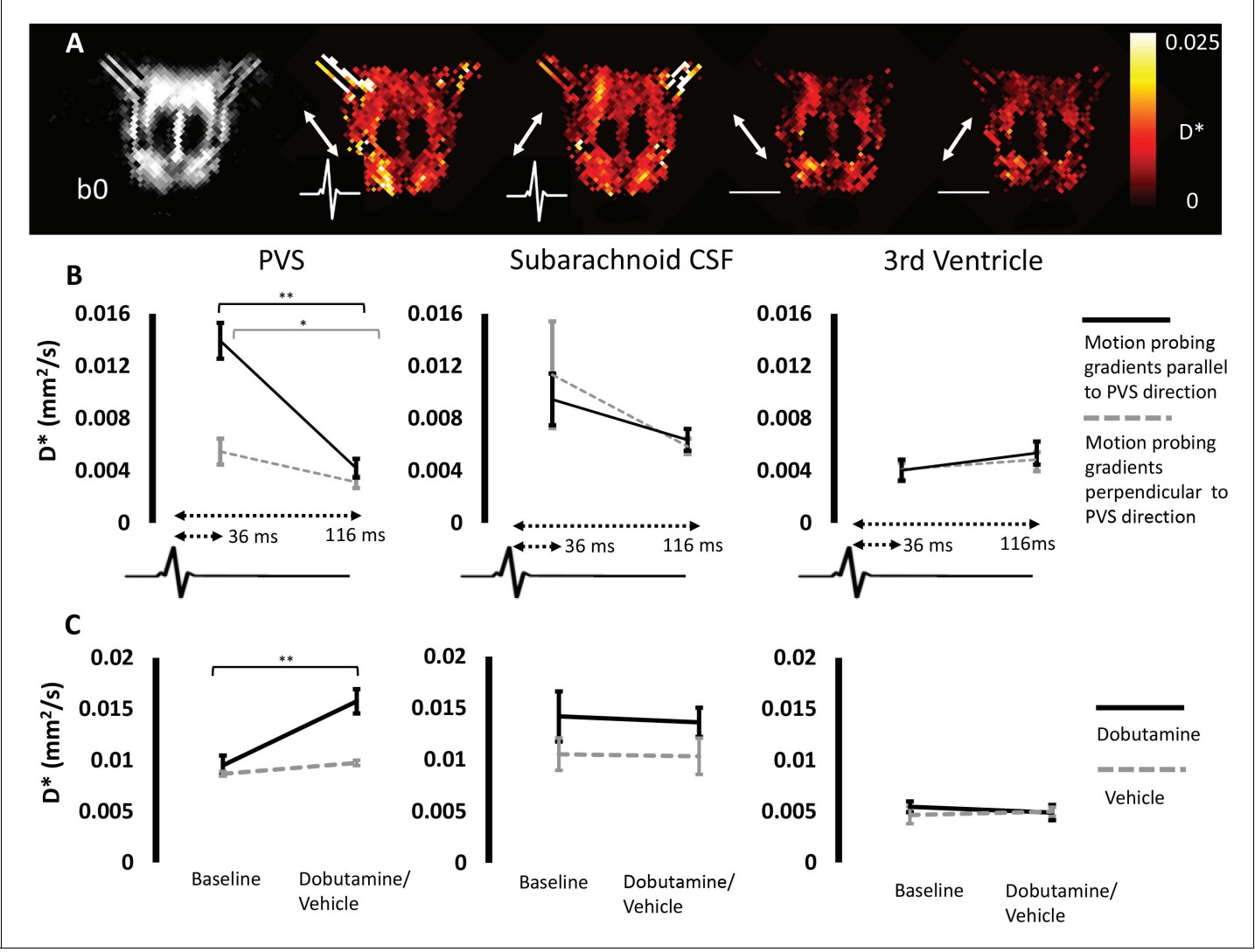

**Figure 4.** Cerebral arterial pulsation drives non-invasive measures of perivascular fluid movement. (**A**) b0 image (first column) and D* maps during arterial pulsation (second and third column) and during diastole (fourth and fifth column) from a single animal [the white arrows represent the direction of the applied MPGs]. (**B**) The mean D* during arterial pulsation and diastole respectively within the three ROIs for MPGs applied parallel (black line) and perpendicular to (grey dashed line) PVS orientation. (**C**) The mean D* at baseline and after dobutamine (black line) or vehicle (grey dashed line) within the same ROIs (non-gated).

DOI: https://doi.org/10.7554/eLife.34028.008

The following figure supplement is available for figure 4:

**Figure supplement 1.**

DOI: https://doi.org/10.7554/eLife.34028.009

signal and hence that the parameters extracted from their measurements provide meaningful correlates of PVS fluid movement.

The expression of AQP4 appears to be mechanistically important in CSF-ISF exchange (*Smith et al., 2017*; *Mestre, 2017*). However, although genetic deletion of AQP4 was found to markedly decrease rates of small molecular weight tracer inflow from the CSF into the brain, it did not appear to affect the movement of tracers along para-arterial channels (*Iliff et al., 2012*). Thus, by extension, as the technique here is targeted to PVS fluid movement, it may not be sensitive to AQP4 related modulation of CSF-ISF exchange through genetic deletion of AQP4 in the rodent brain. Hence, future studies are required to fully elucidate the relationship between para/perivascular fluid movement, CSF-ISF exchange and AQP4 expression (*Brinker et al., 2014*). Furthermore,

rates of glymphatic inflow have been linked to changes in extracellular space volume (*Xie et al., 2013*) and central noradrenaline activity (*Benveniste et al., 2017*) and how these factors may modulate measures of D* captured using the techniques presented here would be an interesting avenue of further study. Moreover, how the technique introduced here may be influenced by pathology is an important consideration. For example, the composition of the CSF and PV fluid may change in disease, in turn altering the relaxation times of this compartment (for example the presence of iron could reduce PVS T2). Whilst this may not confound measures of D*, as relaxation time changes will be accounted for by the acquisition of a b0 image at identical TR and TE, this may change contrast between the PVS and surrounding tissue and potentially lessen the SNR of the measurements. MR relaxometry studies targeted to the normal and enlarged PVS may be an interesting avenue of future investigation leading to novel biomarkers of PVS composition. Efforts are ongoing to investigate the sensitivity of the method to detect dysfunction of perivascular fluid movement associated with ageing and models of pathological conditions, with the knowledge that clinical translation of this non-invasive approach may be practically achievable in the near future.

## Materials and methods

All experiments were performed in accordance with the UK Home Office's Animals (Scientific Procedures) Act (1986). In total, 27 male Sprague Dawley rats were used in these experiments (n = 10 for multi-direction diffusion weighted imaging, n = 6 for diffusion tensor imaging, n = 5 for ECG gating, n = 6 for dobutamine). Anaesthesia was induced using 4% isoflurane in 0.4 L/min medical air and 0.1 L/min $O_2$ and was maintained at 2% isoflurane whilst the animal was placed on a MRI compatible plastic probe. The head was secured using ear bars, a bite bar and a nose cone to minimise motion during the data acquisition. Once the probe was fixed in the scanner, isoflurane concentration was reduced to 1.75% in 0.4 L/min medical air and 0.1 L/min $O_2$. Core body temperature was measured throughout using a rectal thermometer (Small Animal Instruments Inc.) and maintained at $37 \pm 0.5°C$ using heated water tubing during the preparation and heated water tubing and warm air flow during the data acquisition period. Breathing rate was monitored throughout the acquisitions using a respiration pillow sensor (Small Animal Instruments Inc.). A scavenger pump was fixed inside the magnet bore to prevent build-up of isoflurane. For the multiple direction diffusion weighting, power analysis based on pilot data was used to estimate the number of animal required to detect a significant difference in D* when motion probing gradients are applied parallel to the perivascular tracts relative to D* when motion probing gradients are applied perpendicular to the perivascular tracts (assuming a normal distribution).

### Magnetic resonance imaging

All imaging was performed using a 9.4T VNMRS horizontal bore scanner (Agilent Inc., Palo Alto, CA). A 72 mm inner diameter volume coil was used for RF transmission and signal was received using a 4 channel array head coil (Rapid Biomedical). The imaging gradient hardware was calibrated using a custom designed structural phantom, as previously described (*O'Callaghan et al., 2014*).

A key aspect of the MRI sequence was the use of a long TE to attenuate the signal from the surrounding arterial blood and tissue (T2 ~30 and 38 ms respectively at 9.4T [*Wells et al., 2013*]) relative to the MRI signal from CSF in the subarachnoid space and fluid in perivascular channels (T2 ~111 ms [*Kuo et al., 2005*]). In order to achieve this, a fast spin echo (FSE) sequence was employed (180° refocusing pulses) with an echo train length of 16 giving an effective echo time of 142 ms (thus the ultra-long TE is compatible with a multiple echo train FSE readout for SNR efficiency). Therefore, at this echo time, the signal from the grey matter tissue, blood and CSF will have decayed to ~2, 1 and 28% of the theoretical signal at TE = 0 respectively. In addition, the use of an ultra-long TE permits a long echo train per excitation (16 echoes) to increase the SNR efficiency of the acquisition (i.e. SNR per unit time). Finally, the use of a relatively long TR (5000 ms), further weights the measured MRI signal from CSF/interstitial fluid relative to surrounding blood/tissue. It should be noted that, as part of the FSE readout, phase encoding lines will be acquired at a range of different TEs and thus the eventual contrast in the image may deviate from that predicted by assuming a constant $TE_{eff}$ across all phase-encoding steps. Simulations (data not shown) indicate that this effect was minimal in the current study but future applications should consider this aspect of MRI image capture.

In this study, four separate sets of experiments were performed, that can be divided into 'multiple direction diffusion weighted imaging' (n = 10), 'diffusion tensor imaging' (n = 6), 'ECG-gating (n = 5)' and' Dobutamine (n = 6)'.

## Multiple direction diffusion weighted imaging (n = 10)

An axial slice was positioned at the ventral aspect of the brain at the level of the Circle of Willis (CoW - see *Figure 1*). A series of scout images were acquired with the slice orientation and position manually altered in an iterative manner in order that the perivascular space around the MCA could be optimally visualised.

The angular orientation of the image was then changed so that the animals right perivascular tracts (surrounding the MCA in the axial slice) was aligned with the orientation of the frequency encoding (FE) imaging gradients. In doing so, the animals left perivascular tracts then become approximately aligned with the phase encoding (PE) imaging gradients (see *Figure 2*). This ensured that, when applying diffusion (or motion probing) gradients along the FE direction, the direction of diffusion weighting was parallel to the right perivascular tract and perpendicular to the left tract; and vice versa when applying diffusion gradients along the PE direction. As a result, the sensitivity for measuring differences in fluid movement along and across both tracts was maximised.

A fast-spin echo imaging sequence was acquired with the following sequence parameters: TR = 5 s, Echo Train Length = 16, effective TE = 142 ms, echo spacing = 10 ms, FOV = 25 $\times$ 25 mm, matrix size = 128 $\times$ 128, slice thickness = 0.8 mm or 1 mm, number of averages = 12. A b = 0 image was acquired with minimal diffusion weighting (b0) and then with separate acquisitions with the motion probing gradients applied in three principle directions (X, Y, Z) with a *b*-value of 107 s/mm$^2$ ($\delta$ = 5 ms, $\Delta$ = 26 ms, G = 4.2 G/cm).

Regions of interest were manually drawn around the perivascular tracts surrounding the left and right MCA, as well as within the CSF of the subarachnoid space in the mid-section of the CoW from the b0 images. The subarachnoid space ROI was chosen because previous invasive measures have demonstrated rapid caudal-rostral CSF-tracer movement in this region (*Gaberel et al., 2014*; *Mesquita et al., 2015*). As such, data from this ROI can provide a degree of validation for the technique if the directionality of fluid movement is found to be consistent with the established caudal-rostral fluid movement. The pseudo-diffusion coefficient (D*) was then calculated for each direction of the applied motion probing gradients using the following equation:

$$S = S0 \exp(-b D^*)$$

where S is the measured signal at $b$ = 107 s/mm$^2$, S0 is the signal taken from the b0 image. In this work we choose to report the exponential decay coefficient as the pseudo diffusion coefficient (D*) since this is analogous to the Intra-voxel Incoherent Motion (IVIM) literature where in-vivo D* estimates reflect an unknown contribution from relatively coherent flow in large and/or directionally ordered vessels and isotropic fluid motion derived from randomly orientated vessels within a MRI voxel.

A paired t-test was applied to investigate (i) if D* was greater when the motion probing gradient was applied parallel to the principle direction of the perivascular tracts, relative to application in each of the orthogonal planes for the left and right perivascular channels respectively; (ii) if D* in the subarachnoid space ROI was significantly greater in the FE and PE directions than in the through plane slice selection direction.

## Diffusion tensor imaging (n = 6)

Images were acquired with no 'diffusion weighting' (b0) and then using motion probing gradients applied in 6 different directions ($\delta$ = 7.5 ms, $\Delta$ = 52 ms, G = 1.5 G/cm, b value ~100 s/mm$^2$) respectively with the following sequence parameters: TR = 5 s, Echo Train Length = 16, effective TE = 142 ms, FOV = 30$\times$15 mm, matrix size = 128$\times$64, slice thickness = 1 mm, number of averages = 24.

Pseudo-Diffusion tensors were generated using a calculated *b*-matrix that incorporated the 'diffusion' weighting introduced by the imaging gradients. As described above, ROIs were drawn around the perivascular tracts that surround the animal's left and right MCA, as well as the CSF in the subarachnoid space that resides in the mid-section of the Circle of Willis. For visualisation purposes, pseudo-diffusion tensor ellipsoids were generated using the fanDTasia routines in Matlab

(*Barmpoutis et al., 2007*). For pseudo-diffusion tensor mapping, images were smoothed using an edge preserving filter and thresholded based on signal intensity, to remove signals that did not principally derive from fluid filled compartments and images were generated using the Explore DTI toolbox (*Leemans et al., 2009*). Maps were colour coded according to their principle orientation. In one animal, the diffusion tensor sequence was applied to the brain immediately post-mortem.

### ECG gating (n = 5)

In these experiments, a three lead electrode was used to measure ECG signals in the bore of the magnet. The diffusion weighted sequence was acquired with the following parameters: TR = 5 s, Echo Train Length = 16, effective TE = 142 ms, echo spacing = 10 ms, FOV = 25 $\times$ 25 mm, matrix size = 128 $\times$ 128, slice thickness = 1 mm, number of averages = 12, $\delta$ = 5 ms, $\Delta$ = 26 ms, diffusion gradient amplitude = 2.3 G/cm, b value ~45 s/mm$^2$, diffusion gradients applied in two directions (in plane, parallel to the PVS around the left and right MCA respectively).

Image capture was gated to the ECG signal and image acquisition began either directly after the r-wave or with an 80 ms delay. Given that the diffusion weighting is applied during the first echo time at 72 ms, this results in a delay of 36 ms from the r-wave to the centre of diffusion weighting (i.e. the first 180° refocusing pulse) or 116 ms with the additional 80 ms delay. As $\Delta$ was 26 ms in these acquisitions, the 'diffusion weighting' was therefore applied between 23 and 49 ms from the r-wave and 103 and 129 ms from the r-wave respectively. Given previous recordings of pulse wave velocity in the mouse brain of 2.69 m/s (*Di Lascio et al., 2014*) and given an approximate distance from the heart to the MCA of 10 cm in ~400 g rats (together with the separation between adjacent r-waves to be ~150 ms) we define the separate acquisitions to therefore take place during cerebral arterial pulsation or diastole. It should be noted that due to the ECG gating employed in these experiments, the TR will vary slightly between successive echo trains, but given the minimum TR was 5 s and that the r-r interval in the rat is ~150 ms, this should introduce relatively little variation into the measured MRI signal. ROIs were drawn around the left and right PVS and within the mid-section of the subarachnoid space as before. In addition, ROIS were drawn within the third ventricle to examine the r-wave delay dependence on measures of D* within ventricular CSF. The average D* in the PVS (MPGs applied parallel and perpendicular to PVS orientation respectively) was taken for each rat and a paired t-test was used to investigate if D* (MPGs parallel to PVS orientation) was greater during arterial pulsation relative to diastole for each region.

### Dobutamine (n = 6)

Data were acquired in 6 male Sprague Dawley rats using the identical MRI sequence approach described above ('ECG gating') but with no ECG gating. Dobutamine (n = 3 subcutaneous bolus, 2 mg/kg (*Buttrick et al., 1988*) in saline ~0.6–0.8 ml) or saline vehicle (n = 3) was then delivered and the same acquisitions were performed after bolus infusion.

## Acknowledgements

JW is supported by the Wellcome Trust/Royal Society (204624/Z/16/Z). DLT is supported by the UCL Leonard Wolfson Experimental Neurology Centre (PR/ylr/18575). This work is supported by the EPSRC-funded UCL Centre for Doctoral Training in Medical Imaging (EP/L016478/1) and the Department of Health's NIHR-funded Biomedical Research Centre at University College London Hospitals. ML receives funding from the EPSRC (EP/N034864/1); the King's College London and UCL Comprehensive Cancer Imaging Centre CR-UK and EPSRC, in association with the MRC and DoH (England); UK Regenerative Medicine Platform Safety Hub (MRC: MR/K026739/1).

## Additional information

### Funding

| Funder | Grant reference number | Author |
|---|---|---|
| Wellcome | Sir Henry Dale Fellowship 204624/Z/16/Z | Phoebe G Evans Jack A Wells |

| Royal Society | Sir Henry Dale Fellowship 204624/Z/16/Z | Phoebe G Evans Jack A Wells |
|---|---|---|
| Engineering and Physical Sciences Research Council | EP/N034864/1 | Ian F Harrison David L Thomas Mark F Lythgoe |
| National Institute for Health Research | | Mark F Lythgoe |
| Medical Research Council | MR/K026739/1 | Mark F Lythgoe |
| Department of Health | | Mark F Lythgoe |
| Leonard Wolfson Experimental Neurology Centre | PR/ylr/18575 | David L Thomas |
| Engineering and Physical Sciences Research Council | UCL Centre for Doctoral Training in Medical Imaging (EP/L016478/1 | Yolanda Ohene Payam Nahavandi |

The funders had no role in study design, data collection and interpretation, or the decision to submit the work for publication.

## Author contributions

Ian F Harrison, Conceptualization, Supervision, Investigation, Writing—review and editing; Bernard Siow, Data curation, Formal analysis; Aisha B Akilo, Phoebe G Evans, Investigation, Methodology; Ozama Ismail, Yolanda Ohene, Payam Nahavandi, Writing—review and editing; David L Thomas, Conceptualization, Data curation, Methodology, Writing—review and editing; Mark F Lythgoe, Conceptualization, Funding acquisition, Methodology, Writing—review and editing; Jack A Wells, Conceptualization, Resources, Formal analysis, Supervision, Funding acquisition, Investigation, Visualization, Methodology, Writing—original draft, Project administration, Writing—review and editing

## Author ORCIDs

Ian F Harrison (ID) http://orcid.org/0000-0003-1250-4911
David L Thomas (ID) http://orcid.org/0000-0003-1491-1641
Jack A Wells (ID) http://orcid.org/0000-0002-4171-3539

## Ethics

Animal experimentation: All experiments were performed in accordance with the UK Home Office's Animals (Scientific Procedures) Act (1986). All procedures were minimally invasive and with a relatively high level of isoflurane for deep anesthesia throughout imaging.

## Decision letter and Author response

Decision letter https://doi.org/10.7554/eLife.34028.015
Author response https://doi.org/10.7554/eLife.34028.016

# Additional files

## Supplementary files

• Transparent reporting form
DOI: https://doi.org/10.7554/eLife.34028.010

## Data availability

All the data has been deposited on Dryad (https://dx.doi.org/10.5061/dryad.121hs31).

The following dataset was generated:

| Author(s) | Year | Dataset title | Dataset URL | Database, license, and accessibility information |
|---|---|---|---|---|
| Harrison I, Siow B, Akilo A, Evans P, Ismail O, Ohene Y, Nahavandi P, Thomas D, Lythgoe M, Wells J | 2018 | Data from: Non-invasive imaging of CSF-mediated brain clearance pathways via assessment of perivascular fluid movement with diffusion tensor MRI | https://dx.doi.org/10.5061/dryad.121hs31 | Available at Dryad Digital Repository under a CC0 Public Domain Dedication |

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
