## [Decision Letter]

Thank you for submitting your article "Non-invasive imaging of CSF-mediated brain clearance pathways via assessment of perivascular fluid movement with DTI MRI" for consideration by *eLife*. Your article has been reviewed by three peer reviewers, and the evaluation has been overseen by a Reviewing Editor and Sabine Kastner as the Senior Editor. The following individuals involved in review of your submission have agreed to reveal their identity: Matthias van Osch (Reviewer #2); Maiken Nedergaard (Reviewer #3).

The reviewers have discussed the reviews with one another and the Reviewing Editor has drafted this decision to help you prepare a revised submission.

Summary:

In this paper the authors discuss a novel non-invasive MRI-based technique for studying the flow of fluid in perivascular channels using a long echo time diffusion-weighted approach. They show data consistent with the motion of fluid in the expected directions along channels close to the circle of Willis in the rat brain. This method could potentially be used to study disruption of the flow of fluid along perivascular channels, with implications for the functioning of the glymphatics system. This is especially pertinent with respect to the recent interest in the clearing capacity of various substances (e.g. amyloid) from the brain in diseases such as in Alzheimer's and cerebral amyloid angiopathy. This could make it an important tool in studying a range of neurodegenerative diseases: its non-invasive nature would make it suitable for use in longitudinal studies, both in animal models and potentially in living humans.

The reviewers felt that generally the paper is well written and discusses an interesting idea with significant potential for studying this system, which has previously only been observed directly using invasive methods. However, the reviewers have a number of concerns that should be addressed prior to publication. Please note that if addressing these points will require taking the manuscript considerably beyond the 2,000 words allowed for a short report then the authors should request that the manuscript type be changed to a research article.

Essential revisions:

1) There is some concern about whether the proposed MRI-based metrics may be sensitive to changes other than glymphatic function (e.g. differences in cardiac pulsatility) which could potentially confound the measurement. It would be useful to see some additional data showing the dependence of the signal on the cardiac (and ideally also respiratory) cycles to help the reader interpret the data being presented here.

2) The claim that the derived diffusion metrics relate to glymphatic function would be significantly strengthened by the addition of experimental data showing an example of where these metrics are modified. This could be through a negative example such as a cisterna magna puncture or AQP4 inhibition. It would also be beneficial to combine the diffusion-based measurements with contrast-enhanced MRI to anatomically identify the perivascular space of the MCA, and thereby confirm the anatomical location. If the authors feel such experiments are not possible within the revision period, then additional discussion of signal interpretation, potential confounding effects and future experiments that could help confirm the sensitivity of the proposed method to glymphatic function would be required.

3) It would be informative for the authors to discuss how this method would perform in pathology. Especially, whether the T2 of CSF in perivascular spaces would be lower due to increased level of waste products including iron. Furthermore, the authors leave it quite open how they would apply this technique to study pathology. Would they focus more on FA or on the apparent diffusion coefficients? It feels that the current approach focuses a lot on the orientation of CSF-flow, which is perhaps only interesting as proof-of-concept evidence. Moreover, one might argue that fluctuations in CSF mobility in the perivascular spaces would be even more interesting than the mean diffusivity. A more elaborate discussion on this topic would be worthwhile.

---

## [Author Response]

Essential revisions:1) There is some concern about whether the proposed MRI-based metrics may be sensitive to changes other than glymphatic function (e.g. differences in cardiac pulsatility) which could potentially confound the measurement. It would be useful to see some additional data showing the dependence of the signal on the cardiac (and ideally also respiratory) cycles to help the reader interpret the data being presented here.

This is an excellent suggestion, especially given evidence from earlier studies that ‘cerebral arterial pulsatility is a key driver of paravascular CSF influx’ (Iliff et al.,J Neurosci, 2013) or ‘This apparent convective tracer influx may be facilitated by transmission of the pulsations of the cerebral arteries to the microvasculature’ (Rennels et al., 1985). Thus, cardiac pulsation is expected to be an important mechanism responsible for fluid movement in the PVS.

Additional experiments (n=5) were performed with the acquisition gated to the heart beat using ECG signal measured in the MRI scanner. Images were captured during cerebral arterial pulsation (36ms delay from the r-wave to centre of ‘diffusion weighting’) and diastole (116ms delay from r-wave). The results are shown in Figure 4A, which demonstrates that arterial pulsatility is indeed a prominent driver of perivascular fluid movement and that we can capture this mechanism using the non-invasive techniques introduced here. Figure 4A shows maps of D*, where the PVS can be seen to ‘light up’ during arterial pulsation in a highly directional manner. D* along the PVS was found to be ~300% greater during arterial pulsation relative to diastole. These data suggest that PVS fluid movement is distinctly dependent on cerebrovascular pulsations relative to subarachnoid/ventricular CSF (which showed more subtle dependence on arterial pulsatility). Furthermore, as suggested by the reviewer, we performed the same measurements but gated to the respiratory cycle in a single animal (acquiring during the ‘inhale’ and resting period (non-ECG gated)). In doing so, D* (parallel to PVS orientation) was found to be highly similar [D*=0.0094 mm^2^/s (inhale) and 0.0086 mm^2^/s (resting period)] suggesting that the respiratory cycle has a very minor role in driving PV fluid movement relative to cerebral vascular pulsation.

This finding was robust across the 5 rats, where a highly significant dependence on arterial pulsatility was observed in the PVS (p=0.002, Figure 4). Conversely no differences in D* due to the ECG-gated delay (36 or 116ms) were observed in the 3^rd^ ventricle (=0.4). We thank the reviewers for their suggestion which has led to an exciting result – the technique highlights the movement of fluid in the PVS as functionally distinct from subarachnoid and ventricular CSF, given the marked vascular pulsatility dependence observed (Figure 4). As discussed in the revised manuscript, this suggests that PVS fluid movement, as an enabling pathway for CSF-ISF exchange, is uniquely tied to vascular pulsatility and that we can capture this mechanistic link non-invasively. These new data have been incorporated into the revised manuscript and are presented in full within a new Figure (Figure 4). All the individual animal data the make up Figure 4 are shown in Figure 4—figure supplement 1.

2) The claim that the derived diffusion metrics relate to glymphatic function would be significantly strengthened by the addition of experimental data showing an example of where these metrics are modified. This could be through a negative example such as a cisterna magna puncture or AQP4 inhibition.

Following the reviewers’ comment, we sought to investigate whether the ‘diffusion’ metrics captured here would positively correlate to the increase in perivascular influx associated with dobutamine administration (Iliff et al., J.Neuroscience 2013). Additional experiments were performed in 6 rats with baseline measures (non-gated) taken and then repeated following administration of dobutamine (n=3) or vehicle (n=3). The results are shown in new Figure 4C, where a marked increase in D* along the PVS was recorded following dobutamine relative to vehicle. These new data are included in Figure 4 and, in addition, we now discuss these new findings and future experiments that would help further characterise sensitivity of the method to glymphatic function:

Discussion: **“**The precise mechanisms that underlie CSF-ISF exchange are yet to be fully defined and this remains an active area of research. Accumulative evidence, however, has established cerebral vascular pulsation as an important mechanism underlying perivascular fluid movement (Rennels et al., 1985; Iliff et al., 2013). […] The non-invasive nature of this technique may enable future studies to investigate the mechanistic link between vascular pulsatility and PVS fluid movement in the healthy human brain, and its modulation by pathology as well as novel therapy.”

Discussion: **“**The expression of AQP4 appears to be mechanistically important in CSF-ISF exchange (Smith et al., 2017; Mestre et al., 2017). […] Furthermore, rates of glymphatic inflow have been linked to changes in extracellular space volume (Xie et al., 2013) and central norepinephrine activity (Benveniste et al., 2017) and how these factors may modulate measures of D* captured using the techniques presented here would be an interesting avenue of further study.”

It would also be beneficial to combine the diffusion-based measurements with contrast-enhanced MRI to anatomically identify the perivascular space of the MCA, and thereby confirm the anatomical location. If the authors feel such experiments are not possible within the revision period, then additional discussion of signal interpretation, potential confounding effects and future experiments that could help confirm the sensitivity of the proposed method to glymphatic function would be required.

Following the reviewer’s comment, we have sought to compare the non-invasive approach introduced here with the more established contrast enhanced MRI technique. Author response image 1 shows dynamic T1-weighted data acquired using identical apparatus in our lab (male Sprague Dawley rat, 300g) following cisterna magna infusion of gadolinium. At baseline, the morphology of the MCA (bright in T1 weighted image at baseline) can be seen to closely match the orientation of the PV channels imaged using the non-invasive approach. At 12 minutes after gadolinium infusion, the tracer is localised in the tissue immediately surrounding both branches of the MCA (Author response image 1, shown by ‘*’), highlighting the functional role of the PVS to facilitate CSF-ISF exchange. At 24 minutes post-infusion, the tracer has further egressed into the tissue outwards from the MCA (highlighted by the yellow dashed arrows). Thus, Author response image 1 provides further reassurance that we have correctly identified the PVS around the MCA for non-invasive assessment of PVS function in this study.

**Author response image 1. respfig1:** Comparison to contrast enhanced MRI following cisterna magna injection of gadolinium A. b0 image at the ventral aspect of the rat brain captured using the noninvasive approach with an ultra-long TE. B. T1 weighted image at the matched ventral slice at baseline, prior to intra-cisternal gadolinium infusion. T1 weighted images 12 minutes (**C**) and 24 minutes (**D**) after intra-cisternal gadolinium infusion.

3) It would be informative for the authors to discuss how this method would perform in pathology. Especially, whether the T2 of CSF in perivascular spaces would be lower due to increased level of waste products including iron. Furthermore, the authors leave it quite open how they would apply this technique to study pathology. Would they focus more on FA or on the apparent diffusion coefficients? It feels that the current approach focuses a lot on the orientation of CSF-flow, which is perhaps only interesting as proof-of-concept evidence. Moreover, one might argue that fluctuations in CSF mobility in the perivascular spaces would be even more interesting than the mean diffusivity. A more elaborate discussion on this topic would be worthwhile.

Following the reviews comment, we now elaborate on this important issue in the Discussion. We agree with the reviewer that, especially given the new data acquired in Figure 4, that fluctuations in D* measured parallel to the PVS over the cardiac cycle may provide the most promising correlate to pathological disruption of PVS function. However, future studies are required to fully elucidate the DTI metrics derived from the techniques here that give the most meaningful insights into pathological processes.

Discussion: **“**Moreover, how the technique introduced here may be influenced by pathology is an important consideration. […] Efforts are ongoing to investigate the sensitivity of the method to detect dysfunction of perivascular fluid movement associated with ageing and models of pathological conditions.”